# OpenReview forum: "Stochastic Monkeys at Play: Random Augmentations Cheaply Break LLM Safety Alignment"
_ICLR.cc/2025/Conference — ICLR 2025 Conference Withdrawn Submission_

### Official Review · Reviewer_UNaU · 2024-10-29

**Soundness:** 3
**Presentation:** 2
**Contribution:** 3
**Rating:** 5
**Confidence:** 4

**Summary:**

Facing the demand of high-resource and sophisticated design of prompts in jailbreaking attacks on LLM safety alignment, this paper studies how simple random augmentations to the input prompt bypass safety alignment. The authors perform an evaluation of 17 models and investigate random augmentations with multiple dimensions, such as augmentation type, model size, etc.

**Strengths:**

1. The method proposed by the author is simple yet effective. Random augmentation is a simple modification to input prompts, yet such method can increase the success rate of harmful requests by up to 20~26%.
2. The evaluation across multiple dimensions is thorough. The authors investigate random augmentation in augmentation type, model size, quantization, fine-tuning-based defenses, and decoding strategies. And the authors reveal several observations, such as character-level augmentations tend to be more effective than string insertion augmentations.

**Weaknesses:**

1. The authors claim their method to be "cheap", but no experiments support this point. The authors can add computation cost compared to other jailbreaking methods.
2. There is no comparison to other jailbreaking methods. The authors' statement that "adversarial attacks typically assume white-box...random augmentations are black-box...we do not compare..." is not reasonable. There also exist some jailbreaking methods against black-box models. The authors should compare with them. (https://arxiv.org/abs/2310.08419 ; https://arxiv.org/abs/2312.02119)
3. No widely-used commercial LLMs are evaluated. The authors should evaluate the proposed methods against widely-used commercial LLMs (e.g. ChatGPT), where bypassing the safety alignment is a more significant threat.
4. There is no discussion of mitigation methods against the proposed methods.

**Questions:**

1. How safety judge work? Is it done by LLMs, human experts or word matching?

---

> ### Author Response · Authors · 2024-11-29
>
> Thank you for your feedback and questions. We would like to address your concerns:
>
> 1. > “There is no comparison to other jailbreaking methods.”
>     * **Response**: Thank you for your suggestion to compare against other jailbreak methods. We have updated our paper with PAIR and DeepInception results, which can be found in Appendix D.4. A summary of our results can be found in our Global Response (item 4). We have also removed the statement that "adversarial attacks typically assume white-box...random augmentations are black-box...we do not compare..." in our revised draft.
> 2. > “The authors can add computation cost compared to other jailbreaking methods.”
>     * **Response**: Thank you for pointing this out. We have provided measurements of execution time alongside our comparison with other jailbreak methods; please see Appendix D.4 and Global Response item 4.
> 3. > “The authors should evaluate the proposed methods against widely-used commercial LLMs (e.g. ChatGPT), where bypassing the safety alignment is a more significant threat.”
>     * **Response**: We have added GPT-4o evaluations to our paper; please see Global Response item 1 and Appendix D.1 for more details.
> 4. > “There is no discussion of mitigation methods against the proposed methods.”
>     * **Response**: Thank you for this suggestion. We have added discussion on possible mitigation methods in Appendix B (lines 805-814).
> 5. > “How safety judge work? Is it done by LLMs, human experts or word matching?”
>     * **Response**: As mentioned in line 227, we use the safety judge provided by the SORRY-Bench dataset, which is a Mistral LLM fine-tuned for safety classification.

---

> ### Author Response · Authors · 2024-12-01
>
> Thank you again for your initial review. Your comments have been valuable to us for improving our paper.  As the discussion period concludes, we kindly request that you review our recent updates to the paper. In addition, if these updates address your primary concerns, we'd greatly appreciate if you could consider raising your score accordingly.
>
> Warm regards,
>
> Authors

---

> ### Author Response · Authors · 2024-12-03
> **Request for update from Reviewer UNaU**
>
> Dear Reviewer UNaU,
>
> As the rebuttal period comes to a close, we kindly ask you to update your initial review taking into consideration our recent changes. We believe that our paper updates have addressed your core concerns. In particular, we would like to highlight that Reviewer aoVp has updated their score from a 3 to a 6. We greatly appreciate the time you have already put into your review; your engagement matters immensely to us.
>
> Sincerely,
>
> Authors

---

### Official Review · Reviewer_z9us · 2024-10-30

**Soundness:** 2
**Presentation:** 3
**Contribution:** 2
**Rating:** 3
**Confidence:** 4

**Summary:**

The paper investigates jailbreaking open-source LLMs via small adjustments to the prompt. The paper investigates two types of augmentations: a "string" level insertion of a new word, and a "character" level augmentation which edits, deletes, or inserts a character.
The paper evaluates using the SORRY-bench dataset, across a variety of small language models, showing that augmentation can improve ASR under a variety of sampling temepratures. The paper also evaluates the attack under different defenses and under model quantization.

**Strengths:**

The method is simple and shows promise as a new form of jailbreak attack. The writing is clear and easy to understand, and the results are well-presented.

**Weaknesses:**

I was not entirely convinced by the evaluation results; Although they looked promising, it would be much stronger if the paper also included more diverse results.

For example, evaluation with/without system prompts would help clarify limitations of the approach. Additional experiments with larger models, or with instruction-tuned models would also further convince me that the approach is robust and general (or more importantly, reveal something interesting about the nature of LLM robustness to perturbed prompts).

I would also like to see more comparison to prior work; The paper evaluates comparing to (Huang et al 2023), which I felt was interesting as it demonstrates an attack surface that isn't covered by prior work (this point should be emphasized more).

To make this more complete, it would be interesting if the paper compared against the other cited work: Andriushchenko & Flammarion, 2024, Vega et al. 2023, and Zou et al. 2023.

**Questions:**

Restating a few points from Weaknesses:

1) How does your method compare to prior work that adjusts prompts? While there is some evaluation comparing to Huang et al., it would be extremely interesting to build a more complete picture of the attack surface

2) How does your attack perform on instruction-tuned models and when including (or excluding) system prompts? (it's also unclear if system prompts were used during evaluation)

3) How susceptible are closed-source models to this attack?

4) Have you tried string-level augmentations that insert words from a dictionary instead of random text? Would this meaningfully change anything?

5) I could not find any data with varying choices of p. While the paper mentions p=0.05 works well, it would be interesting to see how p changes the strength of the attack.

---

> ### Author Response · Authors · 2024-11-29
>
> Thank you for your detailed feedback and suggestions! We would like to address your concerns:
>
> 1. > “How does your attack perform on instruction-tuned models and when including (or excluding) system prompts? (it's also unclear if system prompts were used during evaluation)”
>     * **Response**: This is an excellent suggestion. We have added experiments using a safety-encouraging system prompt to our paper, which can be found in Appendix D.2. A summary of our results can be found in our Global Response (item 2). Our original submission used the default system prompt for each model, and upon closer inspection, we observed that by default most models leave their system prompt empty, whereas Qwen 2 and Vicuna utilize default system prompts that may impact the results (e.g. “You are a helpful assistant” for Qwen 2). In our revised draft, we have updated the experimental results in the main text so that Qwen 2 and Vicuna have empty system prompts to be consistent with the other models. We have also added a note on line 284 regarding the system prompt to help clarify our setup in the paper. Please see Correction #1 in our Global Response for more details.
> 2. > “Additional experiments with larger models, or with instruction-tuned models would also further  convince me that the approach is robust and general”
>     * **Response**: Thank you for your suggestion. We would like to clarify that all the models we investigate are instruction-tuned, as mentioned in Section 3.2. Specifically, the Llama, Phi and Qwen families have undergone explicit safety alignment, and the remaining families were included to see if any interesting patterns can be observed for unaligned models. For larger models beyond 13B, we provide GPT-4o evaluations; please see Global Response item 1 and Appendix D.1 for more details.
> 3. > “How does your method compare to prior work that adjusts prompts?”
>     * **Response**: Thank you for your suggestion to compare against prior works. We have updated our paper with PAIR and DeepInception results, which can be found in Appendix D.4. A summary of our results can be found in our Global Response (item 4). In terms of literature review, we have also expanded discussion of comparison to prior work that were cited in the introduction; please see Appendix A.1.
> 4. > “How susceptible are closed-source models to this attack?”
>     * **Response**: We have added GPT-4o evaluations to our paper; please see Global Response item 1 and Appendix D.1 for more details. In short, GPT-4o is safer than the other open-source models, but nonetheless we can still find successful jailbreaks (see Figures 7 and 8 in the appendix for examples).
> 5. > “Have you tried string-level augmentations that insert words from a dictionary instead of random text?“
>     * **Response**: This is a great suggestion. However, due to time constraints, we have decided to prioritize other experiments over adding a new augmentation type, and therefore leave this investigation to future work.
> 6. > “I could not find any data with varying choices of p.”
>     * **Response**: Thank you for this suggestion! We have added ablation experiments on p to our paper; please see Appendix D.3 for full details and Global Response item 3 for a summary.

---

> ### Author Response · Authors · 2024-12-01
>
> Thank you again for your initial review. Your comments have been valuable to us for improving our paper.  As the discussion period concludes, we kindly request that you review our recent updates to the paper. In addition, if these updates address your primary concerns, we'd greatly appreciate if you could consider raising your score accordingly.
>
> Warm regards,
>
> Authors

---

> ### Author Response · Authors · 2024-12-03
> **Request for update from Reviewer z9us**
>
> Dear Reviewer z9us,
>
> As the rebuttal period comes to a close, we kindly ask you to update your initial review taking into consideration our recent changes. We believe that our paper updates have addressed your core concerns. In particular, we would like to highlight that Reviewer aoVp has updated their score from a 3 to a 6. We greatly appreciate the time you have already put into your review; your engagement matters immensely to us.
>
> Sincerely,
>
> Authors

---

### Official Review · Reviewer_aoVp · 2024-10-31

**Soundness:** 3
**Presentation:** 3
**Contribution:** 2
**Rating:** 6
**Confidence:** 2

**Summary:**

The paper investigates how simple random input augmentations can bypass safety alignment in state-of-the-art large language models. The authors demonstrate that “stochastic monkeys” (low-resource, unsophisticated attackers) can evade safety mechanisms in LLMs by applying 25 random augmentations to each prompt. Besides, they evaluate these augmentations across several dimensions, such as augmentation type, model size, quantization, fine-tuning-based defenses, and decoding strategies, highlighting that random augmentations enable bypassing of safety features with minimal resources. Notably, character-level augmentations consistently outperform string insertion in breaking alignment. Finally, the paper proposes a novel evaluation metric to evaluate safety and includes a human study to balance false positive and negative rates, furthering the robustness of their findings.

**Strengths:**

1. The paper applies simple random augmentations as a threat model, contributing a fresh perspective to the understanding of LLM safety alignment vulnerability.

2. The study is thorough, evaluating the effectiveness of random augmentations across multiple dimensions, and the use of a human-validated metric strengthens the reliability of the findings.

3. By demonstrating that low-cost, unsophisticated methods can bypass LLM safety features, the paper reveals critical vulnerabilities that may shape future research in secure LLM deployment, particularly for sensitive or public-facing applications.

**Weaknesses:**

1. While the authors discuss their method’s simplicity relative to adversarial attacks, a more detailed comparison against other established jailbreak and adversarial attack methods would better contextualize the effectiveness of random augmentations.

2. The study could benefit from an expanded discussion on the practical implications of these findings, especially in terms of what types of applications or LLM deployment scenarios are most at risk and possible mitigations.

3. The quantization experiment results in Figure 4 show variability across different models and sizes, but the authors provides vague explanations for these differences. Further exploring why certain models (e.g., Qwen 2) exhibit unique behavior under augmentations would enhance the study.

4. The writing of the conclusion is inconsistent with the abstract and conclusion, making it difficult to understand the main takeaway of this paper.

**Questions:**

1. Can the authors provide a comparison of the success rate of random augmentations against the success rate of more sophisticated adversarial attacks in the same LLM models?

2. Among all these various dimensions( augmentation type, model size, quantization, fine-tuning-based defenses, and decoding strategies), can authors provide a ranking of these factors based on their significance towards the attack performance? In other words, the reader would like to know which factors should be the top concerns in their LLM applications.

3. Have the authors considered examining the temporal robustness of this attack, i.e., does the attack success vary over repeated attempts or prolonged interaction with the same model?

---

> ### Author Response · Authors · 2024-11-29
>
> Thank you for your detailed feedback and useful suggestions. We would like to address your concerns:
>
> 1. > “Can the authors provide a comparison of the success rate of random augmentations against the success rate of more sophisticated adversarial attacks in the same LLM models?”
>     * **Response**: Thank you for your suggestion to compare against existing attacks. We have updated our paper with PAIR and DeepInception results, which can be found in Appendix D.4. A summary of our results can be found in our Global Response (item 4).
> 2. > “The study could benefit from an expanded discussion on the practical implications of these findings, especially in terms of what types of applications or LLM deployment scenarios are most at risk and possible mitigations.”
>     * **Response**: This is a great suggestion. We have added much expanded discussion on assessing risk and possible mitigation strategies in Appendix B.
> 3. > “Among all these various dimensions( augmentation type, model size, quantization, fine-tuning-based defenses, and decoding strategies), can authors provide a ranking of these factors based on their significance towards the attack performance?”
>     * **Response**: Certainly! We have updated our paper with a ranking in Section 4.4.
> 4. > “The quantization experiment results in Figure 4 show variability across different models and sizes, but the authors provides vague explanations for these differences. Further exploring why certain models (e.g., Qwen 2) exhibit unique behavior under augmentations would enhance the study.”
>     * **Response**: We agree that Qwen 2 requires a deeper exploration to more fully explain its results in the quantization experiment. On the surface, the reason that W4A16 attains a negative success rate gain is that the response quality is poor, as shown in Figure 19. This of course begs the question: why is the response quality so poor for W4A16 Qwen 2 specifically? One possible explanation is from the recently proposed scaling laws for precision [1], which observed that increasing the pre-training data size incurs greater performance degradation after post-training quantization. Qwen 2 7B was trained on 7 trillion (T) tokens [2], which is greater than Llama 2 (2T) [3] and Phi 3 (4.8T) [4]. (The pre-training data size for Mistral is not publicly available. Also note that Vicuna and Zephyr are fine-tuned versions of Llama 2 and Mistral, respectively.) However, Llama 3 and Llama 3.1 were trained of 15T tokens [5], yet do not experience such degradation in our experiments. Hence, the proposed scaling law does not completely explain the results for Qwen 2, and thus further investigation is required that may be outside the scope of our work (but nonetheless would be quite an interesting future research direction!)
> 5. > “The writing of the conclusion is inconsistent with the abstract and conclusion, making it difficult to understand the main takeaway of this paper.”
>     * **Response**: Thank you for pointing this out. We have updated our conclusion to make it more consistent with the abstract and to make the main takeaway of the paper more focused.
> 6. > “Have the authors considered examining the temporal robustness of this attack, i.e., does the attack success vary over repeated attempts or prolonged interaction with the same model?”
>     * **Response**: This is a great suggestion! While examining repeated attempts would be interesting to investigate, due to time constraints we have decided to prioritize other additional experiments, and leave this investigation for future work.
>
> [1] Kumar, Tanishq, et al. "Scaling Laws for Precision." arXiv preprint arXiv:2411.04330 (2024).
>
> [2] Yang, An, et al. "Qwen2 technical report." arXiv preprint arXiv:2407.10671 (2024).
>
> [3] Touvron, Hugo, et al. "Llama 2: Open foundation and fine-tuned chat models." arXiv preprint arXiv:2307.09288 (2023).
>
> [4] Abdin, Marah, et al. "Phi-3 technical report: A highly capable language model locally on your phone." arXiv preprint arXiv:2404.14219 (2024).
>
> [5] Dubey, Abhimanyu, et al. "The llama 3 herd of models." arXiv preprint arXiv:2407.21783 (2024).

---

> ### Author Response · Authors · 2024-12-01
>
> Thank you again for your initial review. Your comments have been valuable to us for improving our paper.  As the discussion period concludes, we kindly request that you review our recent updates to the paper. In addition, if these updates address your primary concerns, we'd greatly appreciate if you could consider raising your score accordingly.
>
> Warm regards,
>
> Authors

---

> > ### Comment · Reviewer_aoVp · 2024-12-03
> >
> > Thanks for addressing my concerns. I will raise my rating accordingly.

---

> ### Author Response · Authors · 2024-12-03
>
> Dear Reviewer aoVp,
>
> Thank you for reviewing our changes. We greatly appreciate your time and improvement in score.
>
> Sincerely,
>
> Authors

---

### Official Review · Reviewer_9Lec · 2024-11-03

**Soundness:** 3
**Presentation:** 2
**Contribution:** 2
**Rating:** 5
**Confidence:** 4

**Summary:**

This paper examines the vulnerability of state-of-the-art large language models (LLMs) to random input augmentation attacks. It explores how different augmentation strategies, model configurations, and generation parameters affect attack success rates. The authors focus on two main augmentation types: Character-Level and String Insertion. Experimental results show that Character-Level augmentation yields higher attack success. Larger models demonstrate better resistance in general, though size alone doesn't ensure robustness. Furthermore, aggressive quantization tends to increase susceptibility to attacks, with variable effects across models. The paper also tests a fine-tuning-based defense, finding that it enhances resistance against longer attacks but remains vulnerable to shorter ones, with resistance varying based on attack lengths simulated during fine-tuning. A side effect noted is an increased rejection rate for benign prompts. Through extensive experimentation, the authors conclude that random input augmentation is a low-cost yet effective attack against LLM safety alignment, providing insights for future work in enhancing model robustness.

**Strengths:**

1. The paper is well-organized and clearly written, with a coherent structure and logical progression that supports the arguments effectively.
2. The paper examines the impact of various aspects such as LLM parameter size, quantization strategies, sampling parameters, and fine-tuning on resisting random input augmentation attacks

**Weaknesses:**

1. This paper focused on open-source chat models but lacks research on system prompts in the attack and defense settings
2. This paper did not discuss the impact of different tokenizer vocabulary sizes on this attack

**Questions:**

1. From the attack perspective, the authors did not discuss how the system prompt was configured for the open-source models in the experimental setup. It’s possible they used the default system prompt for each model or left it empty. In fact, setting the system prompt with a defensive focus (e.g., instructing the model to only output safe content) could impact the results. From a defense perspective, the authors should also examine whether using a system prompt as a defense mechanism is effective.

2. The authors should discuss the vocabulary sizes of the different LLMs tested. Larger vocabulary models are more likely to contain "glitch tokens," which may not have been fully trained and could potentially be more vulnerable to the attack methods proposed. Would these models be more susceptible to such attacks due to this factor?

---

> ### Author Response · Authors · 2024-11-29
>
> Thank you for your valuable suggestions and feedback! We would like to address your concerns:
>
> 1. > “The authors did not discuss how the system prompt was configured for the open-source models in the experimental setup”
>     * **Response**: Thank you for pointing this out. Our original submission used the default system prompt for each model. Upon closer inspection, we observed that by default most models leave their system prompt empty, whereas Qwen 2 and Vicuna utilize default system prompts that may impact the results (e.g. “You are a helpful assistant” for Qwen 2). In our revised draft, we have updated the experimental results in the main text so that Qwen 2 and Vicuna have empty system prompts to be consistent with the other models. We have also added a note on line 284 regarding the system prompt to help clarify our setup in the paper. Please see Correction #1 in our Global Response for more details.
> 2. > “The authors should also examine whether using a system prompt as a defense mechanism is effective.”
>     * **Response**: This is an excellent suggestion. We have added experiments using a safety-encouraging system prompt to our paper, which can be found in Appendix D.2. A summary of our results can be found in our Global Response (item 2).
> 3. > “The authors should discuss the vocabulary sizes of the different LLMs tested.”
>     * **Response**: Thank you for providing this suggestion. We examined vocabulary sizes for each of our models to assess whether there is any relationship between vocabulary size and safety against random augmentations. Our conclusion was that there is no strong relationship between the two. For example, the vocabulary sizes of safety-aligned models Llama 2 7B Chat, Phi 3 Small 8K,  Llama 3.1 8B and Qwen 2 7B are 32000, 100352, 128256 and 152064 respectively, yet their average success rate gains are 9.1%, 13.5%, 5.4%, and 14.4% (source: Table 7). Increasing vocabulary size does not seem to induce a meaningful trend in the success rate gain, and thus we conclude that vocabulary size is not an important factor for robustness. We hypothesize that other factors such as the nature of the alignment dataset may matter much more than vocabulary size, and leave a deeper investigation of training data’s impact to future work.

---

> ### Author Response · Authors · 2024-12-01
>
> Thank you again for your initial review. Your comments have been valuable to us for improving our paper.  As the discussion period concludes, we kindly request that you review our recent updates to the paper. In addition, if these updates address your primary concerns, we'd greatly appreciate if you could consider raising your score accordingly.
>
> Warm regards,
>
> Authors

---

> ### Author Response · Authors · 2024-12-03
> **Request for update from Reviewer 9Lec**
>
> Dear Reviewer 9Lec,
>
> As the rebuttal period comes to a close, we kindly ask you to update your initial review taking into consideration our recent changes. We believe that our paper updates have addressed your core concerns. In particular, we would like to highlight that Reviewer aoVp has updated their score from a 3 to a 6. We greatly appreciate the time you have already put into your review; your engagement matters immensely to us.
>
> Sincerely,
>
> Authors

---

### Official Review · Reviewer_Tz62 · 2024-11-04

**Soundness:** 3
**Presentation:** 3
**Contribution:** 2
**Rating:** 5
**Confidence:** 4

**Summary:**

This paper investigates a simple but effective jailbreak attack using random perturbations. The experimental results demonstrate that this attack can significantly improve the chances of bypassing the safety alignment of 17 different models. Additionally, the paper examines how argumentation type, model size, and decoding strategy impact safety measures.

**Strengths:**

1. This paper is well-written with clear problem formulation
2. This paper demonstrate how easy the safety alignment can be bypassed
3. The ablation analysis of the proposed argumentation attack is detailed

**Weaknesses:**

The positioning and technical contribution of this paper seems to be unclear.
1. As a jailbreak attack paper, it lacks comparison with state-of-the-art blackbox jailbreak attacks (such as PAIR and DeepInception) in terms of attack success rate and computational cost. To strengthen the paper's contribution, the authors may need to include comprehensive comparisons with these baseline methods under the same evaluation settings.
2. As an analysis paper, despite being comprehensive, the conclusions are not new to the community. The techniques used (i.e., perturbation) are already widely employed to evaluate LLMs' safety and robustness (e.g., smoothllm). Furthermore, the impacts of model size, quantization, fine-tuning, and generation configurations on LLM safety have been well-studied previously.

**Questions:**

1. How frequently do random augmented prompts affect the semantic meaning of the original malicious prompt? Since the output is binary, could this lead to false safe/unsafe classifications by the evaluator, potentially impacting the experimental results?
2. Have you considered combining random augmentations with existing jailbreak attacks to see if it can lead to better performance?

---

> ### Author Response · Authors · 2024-11-29
>
> Thank you for your suggestions and valuable critique of our work. We would like to address each of your concerns:
>
> 1. > ”Comparison with state-of-the-art blackbox jailbreak attacks (such as PAIR and DeepInception) in terms of attack success rate and computational cost”
>     * **Response**: Thank you for your suggestion to compare against existing attacks. We have updated our paper with PAIR and DeepInception results along with measurements of execution time, which can be found in Appendix D.4. A summary of our results can be found in our Global Response (item 4).
> 2. > “The techniques used (i.e., perturbation) are already widely employed to evaluate LLMs' safety and robustness (e.g., smoothllm).”
>     * **Response**: Thank you for pointing this out. We would like to clarify that prior work that investigates random augmentations in the context of generative language model safety such as SmoothLLM focuses on applying augmentations purely for defense purposes, whereas our work focuses on applying augmentations for attack purposes. We have updated our Related Works section to discuss this in more detail; please see Appendix A.4.
> 3. > “The impacts of model size, quantization, fine-tuning, and generation configurations on LLM safety have been well-studied previously.”
>     * **Response**: We agree that these dimensions have been previously studied in the context of LLM safety. However, the purpose of our work is not to claim that we are the first to explore these. Crucially, much of these works focuses on evaluation against adversarial attacks such as GCG, are strongly limited in the random augmentations they investigate, or only examine notions of safety other than jailbreak vulnerability (such as hallucinations or privacy). Our work contributes to a greater understanding of an underexplored vulnerability in LLM safety alignment and provides much more comprehensive results for safety under random augmentations than any other work currently in the literature. We have updated our Related Works section to compare our work with existing work in each of these dimensions; please see Appendix A.5.
> 4. > “How frequently do random augmented prompts affect the semantic meaning of the original malicious prompt? Since the output is binary, could this lead to false safe/unsafe classifications by the evaluator, potentially impacting the experimental results?”
>     * **Response**: Indeed, random augmentations can occasionally change the semantic meaning of the original prompt. A safety judge takes in both the original prompt and the model response, and a robust safety judge should not classify a response as “unsafe” if it does not actually address the original prompt’s intent. We observed that the safety judge we use is not robust in this sense, and can occasionally produce false positives due to this phenomenon. This is in part what motivated our use of the $\gamma$ threshold in the (k, $\gamma$)-success rate definition and our human study (see Appendix C and Table 2). The experimental results we report in our paper are therefore calibrated to account for such false positives. Due to time constraints, we did not run additional experiments to quantify the frequency of semantic changes, as we feel that our human study addresses the core concern of false classifications in the evaluator and that these additional measurements would just be supplementary.
> 5. > “Have you considered combining random augmentations with existing jailbreak attacks to see if it can lead to better performance?”
>     * **Response**: This is a great suggestion! While combining random augmentations with other jailbreak attacks would certainly be interesting to investigate, due to time constraints we chose to prioritize adding evaluation results for the existing jailbreak attacks PAIR and DeepInception without being combined with random augmentations. We leave exploring the interactive effects of random augmentations with other attacks up to future work.

---

> ### Author Response · Authors · 2024-12-01
>
> Thank you again for your initial review. Your comments have been valuable to us for improving our paper.  As the discussion period concludes, we kindly request that you review our recent updates to the paper. In addition, if these updates address your primary concerns, we'd greatly appreciate if you could consider raising your score accordingly.
>
> Warm regards,
>
> Authors

---

> ### Author Response · Authors · 2024-12-03
> **Request for update from Reviewer Tz62**
>
> Dear Reviewer Tz62,
>
> As the rebuttal period comes to a close, we kindly ask you to update your initial review taking into consideration our recent changes. We believe that our paper updates have addressed your core concerns. In particular, we would like to highlight that Reviewer aoVp has updated their score from a 3 to a 6. We greatly appreciate the time you have already put into your review; your engagement matters immensely to us.
>
> Sincerely,
>
> Authors

---

### Author Response · Authors · 2024-11-29
**Global Response**

Dear reviewers,

Thank you all immensely for your questions and critiques. We have prioritized the following requested items in our revised draft. **Red colored text in our revised draft indicates a new addition** (with the exception of the values in the appendix tables, where the color-coding indicates positive/negative values as was done in the original draft), with the bulk of changes being in the appendix.

1. Evaluation of closed-source model (GPT-4o) (**z9us**, **UNaU**)
    * **Results** (see Appendix D.1 for full details):
      - While GPT-4o is much safer than the open-source models, we are still able to find successful jailbreaks for it using random augmentations, with character deletion being almost two times more successful than the next best augmentation. We provide example jailbreaks in Figures 7 and 8 in the appendix.
2. Systematic investigation of the effects of safety-encouraging system prompt (**9Lec**, **z9us**)
    * **Results** (see Appendix D.2 for full details):
      - A safety-encouraging system prompt (we use the Llama 2 Chat system prompt for all models) can provide some level of protection for all models, but even so, applying random augmentations can still significantly increase the success rate across all models (e.g. by up to ~20% for Qwen 2).
3. Ablation on the augmentation strength p parameter (**z9us**)
    * **Results** (see Appendix D.3 for full details):
      - We observe the following interesting phenomenon: the success rate gains for character-level augmentations tends to increase as the augmentation strength increases, whereas the success rate gains for string insertion augmentations remain mostly unchanged. This further suggests that the safety alignment of LLMs can effectively “ignore” contiguous “noise” that does not impact the tokenization of the original prompt much.
4. Comparison with state-of-the-art black-box jailbreak attacks (PAIR and DeepInception) in terms of attack success and time cost (**Tz62**, **aoVp**, **UnAU**)
    * **Results** (see Appendix D.4 for full details):
      - **Success rate**: On paper, PAIR and DeepInception improve the success rates significantly more than random augmentations. Qualitatively however, the responses of “successful” attacks for PAIR and DeepInception are sometimes less harmful than for random augmentations. See Figure 11 in the appendix for an example. In short, this is because PAIR and DeepInception tend to produce jailbreaks that ”water down” the harmfulness of the original request in order to mitigate refusals, whereas random augmentations tend to maintain the full harmful nature of the original request. Future work can investigate more accurate assessments of these attacks that better take into account the differences in response harmfulness quality. Either way, **note that our goal with random augmentations is not necessarily to beat the state-of-the-art in black box attacks, but rather to investigate an underexplored vulnerability in LLM safety alignment.**
      - **Time cost**: Random augmentations are much faster to execute than PAIR and DeepInception, with DeepInception being more than twice as slow and PAIR being over four times as slow. Moreover, random augmentations elicit much more concise jailbroken responses than PAIR and DeepInception; on average DeepInception induces nearly twice as long target model responses as random augmentations.
5. Provided discussion on practical implications of our findings: risk assessment, possible mitigations, and a ranking of the various factors we explore in terms of how much they impact safety (**aoVp**, **UNaU**)
    * See Section 4.4 and Appendix B.
6. Greatly expanded discussion of related work to better contextualize the contributions of our work (**Tz62**)
    * See updated lines 76-82, 186-189, and Appendix A.
7. Updated the conclusion to more concisely clarify the overall takeaway of the paper (**aoVp**)
    * See Section 5.

These summarize the major reviewer items we have addressed in our paper revision. Please see our individual responses for comments on other reviewer items.

---

> ### Author Response · Authors · 2024-11-29
> **Global Response (cont.)**
>
> Finally, we have made a few corrections to the existing data presented in our work:
>   * **Correction #1**: Updated model size and quantization experiment (RQ2) results for Qwen 2, Llama 3.1 and Vicuna. The update consists of removing the default system prompts for these models so that they are consistent with the other models that have empty default system prompts for this experiment, as well as fixing a bug in the prompt template for quantized Vicuna models. (The Llama 3.1 default system prompt added some date information, and the Qwen 2 and Vicuna default system prompts encouraged the assistant to be helpful or safe.)
>     - **Results**: The new results are similar to the old ones and thus the conclusions for these experiments are unchanged.
>   * **Correction #2**: Temperature sampling experiment (RQ3) results have been updated following discovery of a bug in the output sampling procedure.
>     - **Results**: The conclusion for this experiment has been updated: for only Llama 2, Llama 3 and Phi 3, character deletion further improves the success rate on top of output sampling. For other results, random augmentations tend to decrease the success rate on top of output sampling. Hence, our main conclusion for RQ3 has been qualified to “output sampling and random augmentations can sometimes work together to provide even greater attack effectiveness.” Nonetheless, our findings that random augmentations are effective under greedy decoding remain unchanged, and we believe this becomes key to the successful random augmentation jailbreaks found for GPT-4o.
>
> Sincerely,
>
> Authors

---

### Author Response · Authors · 2024-12-04
**Rebuttal Summary**

Dear Area Chairs,

The major revisions to our paper are summarized under our Global Response. As shown below, we believe we have addressed each reviewer's core concerns. As of writing, 1. **Reviewer aoVp has raised their intial score of 3 to a 6**, and 2. Although we have yet to hear back from the other reviewers, **our efforts to address their concerns in addition to Reviewer aoVP's score raise provides strong reason to believe that the remaining Reviewers would find our revisions in a similarly favorable light**. We therefore believe that our revised submission **should lean towards an acceptance**.

Thank you in advance for reviewing our rebuttal; we are greatly appreciative of your time!

Sincerely,

Authors

---

### Note · Authors · 2024-12-27

**Comment:**

Dear Area Chairs and Reviewers,

Thank you for taking the time to engage with our paper. We have decided to withdraw our submission and will use your feedback to improve our paper.

Sincerely,

Authors

**Withdrawal Confirmation:**

I have read and agree with the venue's withdrawal policy on behalf of myself and my co-authors.